# Properdin oligomers adopt rigid extended conformations supporting function

**Dennis V Pedersen[1], Martin Nors Pedersen[2], Sofia MM Mazarakis[1], Yong Wang[3], Kresten Lindorff-Larsen[3], Lise Arleth[2], Gregers R Andersen[1]\***

[1]Department of Molecular Biology and Genetics, Center for Structural Biology, Aarhus University, Aarhus, Denmark; [2]Structural Biophysics, X-ray and Neutron Science, the Niels Bohr Institute, University of Copenhagen, Copenhagen, Denmark; [3]Linderstrøm-Lang Centre for Protein Science, Department of Biology, University of Copenhagen, Copenhagen, Denmark

**Abstract** Properdin stabilizes convertases formed upon activation of the complement cascade within the immune system. The biological activity of properdin depends on the oligomerization state, but whether properdin oligomers are rigid and how their structure links to function remains unknown. We show by combining electron microscopy and solution scattering, that properdin oligomers adopt extended rigid and well-defined conformations which are well approximated by single models of apparent n-fold rotational symmetry with dimensions of 230–360 Å. Properdin monomers are pretzel-shaped molecules with limited flexibility. In solution, properdin dimers are curved molecules, whereas trimers and tetramers are close to being planar molecules. Structural analysis indicates that simultaneous binding through all binding sites to surface-linked convertases is unlikely for properdin trimer and tetramers. We show that multivalency alone is insufficient for full activity in a cell lysis assay. Hence, the observed rigid extended oligomer structure is an integral component of properdin function.

**\*For correspondence:**
gra@mbg.au.dk

**Competing interests:** The authors declare that no competing interests exist.

## Introduction

The complement system is an essential aspect of innate immunity providing a first line of defense against invading pathogens as well as maintenance of host homeostasis. The complement cascade is activated when circulating pattern recognition molecules recognize molecular patterns on a pathogen, dying host cells, or immune complexes. Activation can initiate through the classical pathway, the lectin pathway, or the alternative pathway (AP) where the AP also provides an amplification loop for the two other pathways (*Bajic et al., 2015*). In all three pathways, labile protein complexes known as C3 and C5 convertases are assembled. These convertases conduct proteolytic cleavage of complement components C3 and C5, respectively, resulting in the generation of opsonins (C3b and iC3b), anaphylatoxins (C3a and C5a), and assembly of the membrane attack complex (reviewed in *Bajic et al., 2015*).

In the AP, the C3-convertase C3bBb is formed when a complex between C3b and the serine protease factor B (FB) is activated by factor D. At a high surface density of C3b, this C3 convertase becomes a C5 convertase. Properdin (FP) is a positive regulator of these convertases. FP is a 53 kDa protein composed of an N-terminal TGF-β binding (TB) domain followed by six thrombospondin type I repeats (TSR1-6). The protein is heavily post-translationally modified carrying one N-linked glycan, four O-linked glycans, and 14–17 C-mannosylated tryptophan residues in the WxxW motifs present in TSR1-6 (*Pedersen et al., 2019a; van den Bos et al., 2019*). FP is produced predominantly by monocytes, T cells, and neutrophils and circulates as oligomers and is primarily found as dimers,

trimers, and tetramers with a 1:2:1 molar distribution in plasma at a concentration of 4–25 µg/mL (*Blatt et al., 2016*; *Pangburn, 1989*). The functions of FP in relation to AP convertases are well established. (1) FP enhances the recruitment of FB to C3b and thereby stimulates proconvertase assembly; (2) FP slows the dissociation of C3bBb 5- to 10-fold; and (3) FP directly competes with factor I resulting in decreased irreversible degradation of C3b to iC3b (*Pedersen et al., 2019a*; *Pedersen et al., 2017*; *DiScipio, 1981*; *Fearon and Austen, 1975*). Other suggested functions of FP are as a C3b-independent pattern recognition molecule capable of triggering the AP (reviewed in *Blatt et al., 2016*) and as a ligand for the NKp46 receptor on innate lymphoid cells (*Narni-Mancinelli et al., 2017*). The importance of FP in innate immunity and homeostasis is demonstrated by individuals with FP deficiency (PD). PD is a rare X-linked disorder, which can be divided into three subtypes: type I (complete lack of FP), type II (1–10% of normal plasma FP level), and type III (normal plasma level but dysfunctional FP). All PD types are characterized by reduced AP activity, resulting in impaired bactericidal activity and increased susceptibility to Neisseria infections and sepsis (*Skattum et al., 2011*).

Classic negative-stain EM (nsEM) studies of FP dimers, trimers, and tetramers revealed that FP oligomers contain compact eye-shaped vertexes connected by thin connecting structures (*Smith et al., 1984*; *Higgins et al., 1995*). Alcorlo and coworkers presented the first 3D reconstruction of the FP eye-shaped vertices in oligomers and 2D classes of the FP-C3bBb convertase complex (*Alcorlo et al., 2013*). Recently, crystal structures *Pedersen et al., 2019a*; *van den Bos et al., 2019* demonstrated that FP oligomers are formed upon interaction of the TB domain and TSR1 from one FP monomer with TSR4, TSR5, and TSR6 from a second FP monomer (*Figure 1A*). In addition, it was established that the binding site for C3b is formed by FP TSR5 in conjunction with a large loop from TSR6 (*Pedersen et al., 2019a*; *van den Bos et al., 2019*; *Pedersen et al., 2017*). Despite prior attempts to analyze FP dimers and trimers with small-angle X-ray scattering (SAXS) and analytical ultracentrifugation (*Sun et al., 2004*), detailed information regarding the structure and the dynamic properties of FP oligomers is missing. Here we present for the first time a structural description of intact oligomeric FP obtained through a combination of nsEM and SAXS. Based on the recent crystal structures of monomeric FP, we are able to annotate all FP domains in monomeric, dimeric, trimeric, and tetrameric FP in EM 2D classes. Pair distance distributions and atomic models based on solution scattering suggest that the FP oligomers are rather rigid in solution despite their very open structure and that their average conformations have cyclic symmetry. In addition, we demonstrate that the defined structure of FP oligomers is crucial for their biological function.

## Results

### The FP monomer is pretzel shaped

Human FP with a C-terminal His-tag was expressed by HEK293F cells and purified by affinity chromatography. The different FP oligomers were subsequently separated by cation exchange and size exclusion chromatography (SEC). As expected, both recombinant and plasma-derived FP eluted in multiple peaks corresponding to the different oligomerization states (*Figure 1—figure supplement 1*). Besides dimeric, trimeric, and tetrameric FP, we also observed a small amount of monomeric FP (FP1) in both recombinant and plasma-derived samples eluting after 14.0 mL corresponding to a molecular weight (MW) of approx. 90 kDa. This apparent MW is significantly larger than the theoretical MW for monomeric FP (53 kDa) due to glycosylations and a non-globular shape but is identical to the observed MW for the monomeric and disease-associated FP variant E244K described previously (*Pedersen et al., 2017*). We were able to purify a significant amount of FP1, which allowed us to obtain nsEM data for a wild-type FP monomer. The resulting 2D classes revealed a flat pretzel-shaped molecule with apparent overall projected dimensions of 95 × 115 Å containing a small and a large ring-shaped structure sharing one edge (*Figure 1B*). Due to a strong preference in the orientation of FP1, we could not obtain a 3D reconstruction of the molecule. We previously described the non-inhibitory nanobody hFPNb1 that binds FP TSR4 within the SCIN stabilized C3bBbFP complex (*Pedersen et al., 2019a*). By analyzing nsEM 2D classes obtained for FP1 in complex with the nanobody hFPNb1 (*Figure 1C*), we could identify with certainty the position of TSR4 and thereafter the remaining domains in the 2D classes of FP1 and nanobody-bound FP1. The small ring of the FP1 molecule corresponds to the well-described FP 'eye' formed by the TB domain, TSR1, TSR5, and

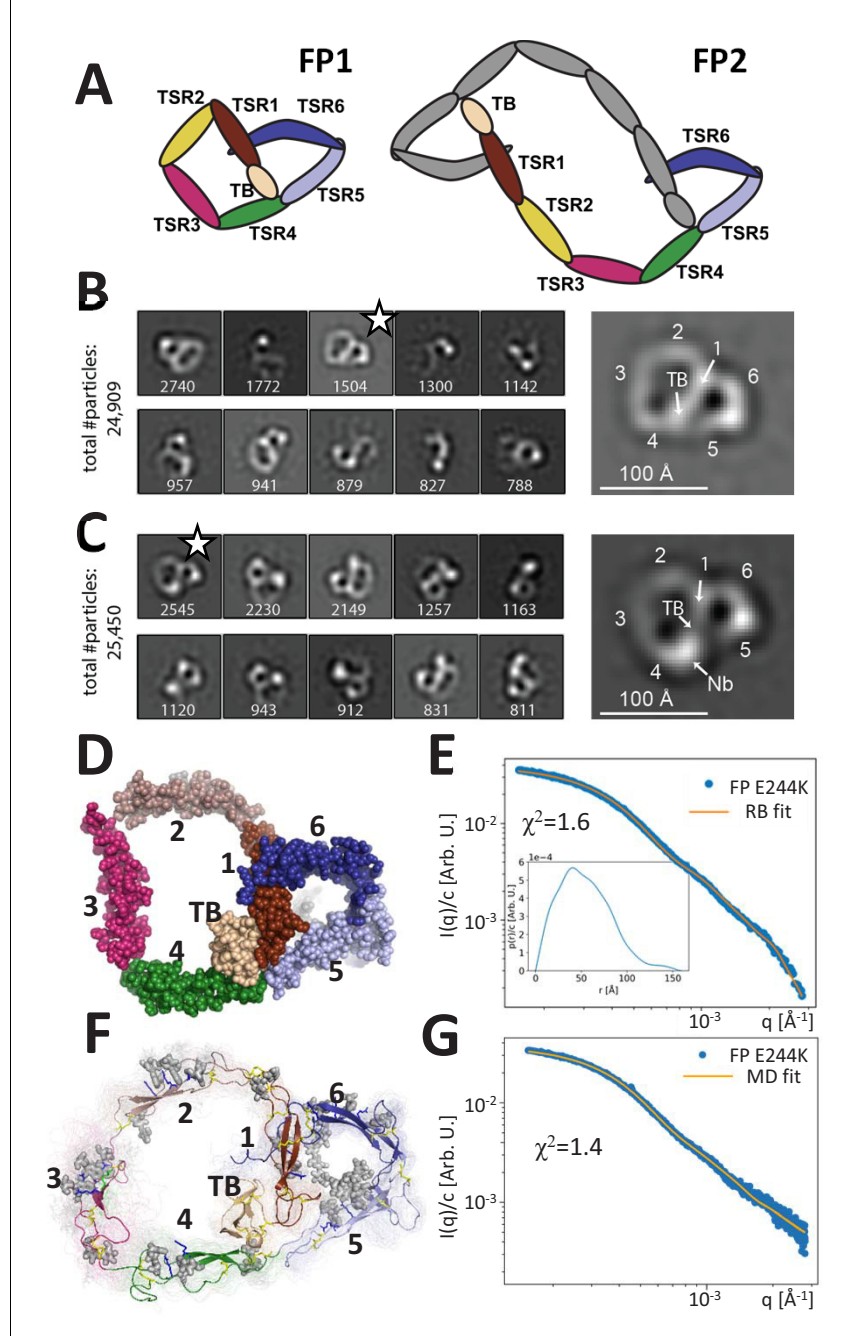

**Figure 1.** Principles of properdin architecture and the structure of FP1. (**A**) Schematic representation of the FP1 monomer and FP2 as an example of an oligomer, where one subunit is colored gray, while the other is colored according to the domain structure as for FP1. (**B**) The 10 most populated nsEM 2D classes obtained with FP1 with the number of particles indicated. A magnified view of the 2D class marked by star is shown to the right. (**C**) As for (**B**), but for the FP1-hFPNb1 complex. Compared to FP1, an additional mass marks the location of hFPNb1 and hence TSR4 enabling assignment of the TB domain and the six thrombospondin repeats in the magnified view to the right. (**D**) Representative atomic model of FP1 E244K derived by rigid-body modeling against the SAXS data. (**E**) Comparison of SAXS experimental data (*Pedersen et al., 2017*) and the fitted scattering curve corresponding to the FP1 E244K model presented in (**E**). Insert to (**E**): $p(r)$ function derived from the SAXS data. (**F**) Conformational ensemble of FP1 E244K sampled by a 1 μs MD simulation represented by 100 frames with 10 ns interval shown as transparent tubes. The starting model is displayed as a cartoon with the glycans and glycosylated residues in gray stick representation. Disulfide bridges are represented by yellow sticks. (**G**) Comparison of SAXS experimental data and the scattering curve obtained from MD ensemble after refinement

*Figure 1 continued on next page*

*Figure 1 continued*

using the Bayesian maximum entropy approach, the two curves fit with $\chi^2$ = 1.4. The minor difference in the experimental data apparent at the highest q-values in (**E**) and (**G**) is due to subtraction of a constant by CORAL in (**E**).

The online version of this article includes the following video and figure supplement(s) for figure 1:

**Figure supplement 1.** Elution profiles of recombinant FP.

**Figure supplement 2.** Molecular dynamics simulation of the FP1 E244K.

**Figure 1—video 1.** Molecular dynamics simulation of FP E244K.

https://elifesciences.org/articles/63356#fig1video1

TSR6 (*Pedersen et al., 2019a*; *van den Bos et al., 2019*), whereas the larger ring is formed by TSR2, TSR3, and TSR4 together with the TB domain and TSR1.

Based on the crystal structure of the recombinant two-chain monomer FPc in which TSR3 and TSR4 are not connected (*Pedersen et al., 2019a*; *Pedersen et al., 2017*), we manually constructed an atomic starting model in accordance with the FP1 2D classes and performed rigid-body refinement of this model against existing SAXS data obtained for the FP1 carrying the E244K mutation (*Pedersen et al., 2017*). Distance restraints secured appropriate distances across the TSR1–TSR2, TSR2–TSR3, and TSR3–TSR4 interfaces and maintained an appropriate distance between the disulfide bridged Cys132 in TSR1 and Cys170 in TSR2. The resulting models clustered tightly and fitted the data with $\chi^2$ = 1.6 (*Figure 1D–E*). The SAXS models strongly resembled the EM 2D classes obtained for the wild-type (WT) FP1 monomer. The $p(r)$ function derived from our published SAXS data on the FP E244K monomer (*Pedersen et al., 2017*) approaches zero at about 130 Å with a small-tail stretching out to a $D_{max}$ at 150 Å (see insert to *Figure 1E*). This agrees with the maximum extent of 140 Å in the nsEM 2D class projections.

To obtain insight into the dynamic properties of FP E244K, we conducted explicit solvent atomistic molecular dynamics simulations of a complete model of FP E244K at microsecond timescale initiating from our SAXS model obtained by rigid-body modeling (*Figure 1—video 1*). The model included the Asn-linked complex glycan, Ser/Thr O-linked glucose-fucose, and the C-linked mannosylations on TSR tryptophans (*Figure 1F*). The SAXS curve calculated from the ensemble of conformations sampled by molecular dynamics simulations is in good agreement with the experimental SAXS data ($\chi^2$ = 2.2). By using experimental data to guide the refinement of the conformational ensemble, the fit could be further improved to $\chi^2$ = 1.4 using the Bayesian maximum entropy (BME) method (*Figure 1G*). Models in this ensemble resembled the FP E244K output models from the SAXS rigid-body refinement (*Figure 1D*) and the 2D classes obtained by EM of wild-type FP1 (*Figure 1B*). Although the mutation E244K is expected to weaken the Trp-Arg stack in TSR3, the Trp-Arg stack remained intact throughout the molecular dynamics (MD) simulation (*Video 1*). However, two independent MD simulations together suggested that TSR3 inserted between TSR2 and TSR4 is capable of rotating as a rigid body with limited conformational flexibility within domain (*Figure 1—figure supplement 2*). This domain rotation was apparently facilitated by high mobility of the TSR4 loop region Asn285-Phe295 facing TSR3, which is in excellent agreement with weak electron density we observed for this region in two crystal structures (*Pedersen et al., 2019a*). At the TSR2–TSR3 junction, the corresponding TSR3 loop Ser225-Pro237 facing TSR2 likewise exhibited flexibility (*Figure 1—figure supplement 2C,D* and *Figure 1—video 1*). For the remaining parts of FP E244K, the simulation did not indicate significant mobility, and in particular the two loops in TSR5 and TSR6 involved in convertase recognition (*Pedersen et al., 2019a*; *van den Bos et al., 2019*) did not exhibit significant changes in backbone conformation (*Figure 1—figure supplement 2C* and *Figure 1—video 1*). Overall, the nsEM 2D classes for WT FP1 and the SAXS data for FP E244K provided the first quasi-atomic experimental model for the monomeric FP1 and FP E244K. Although FP1 like FP E244K probably has limited biological activity, it is present in plasma FP at very low concentrations (*Figure 1—figure supplement 1B*).

## Pseudosymmetric FP oligomer conformations are observed by EM

We next analyzed the properdin dimer (FP2) isolated using SEC by negative-stain EM. Of the 51% of particles that were classified, 8/10 appeared in similar 2D classes, suggesting that a limited number

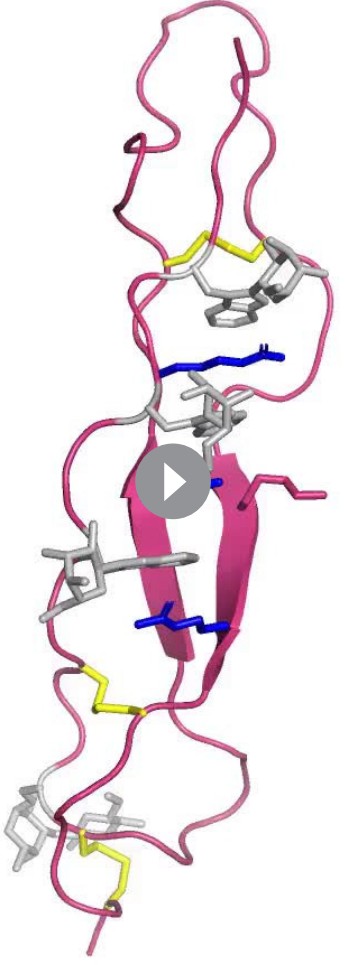

**Video 1.** Close-up on TSR3 during the MD simulation of FP E244K. The thrombospondin repeat is shown in the orientation and coloring used in *Figure 1—figure supplement 2*. Notice how the central Trp-Arg stack remains stable despite the presence of a lysine at position 244. In the structure of wild-type FPc, the glutamate at position 244 engages the side chain of Arg220 in an electrostatic interaction and thereby stabilizes the arginine conformation.

https://elifesciences.org/articles/63356#video1

of related elliptical dimer conformations are present on the EM grids (*Figure 2A*). The overall projected dimension of FP2 in a typical 2D class was 270 × 130 Å with an inner opening of 110 × 50 Å. However, in some classes, the central opening appeared broader and the molecule less extended. At each end of the dimer, the characteristic eye-shaped structure was present and the TSRs could be assigned by comparison with the 2D classes of FP1. Interestingly, TSR3 of FP2 appeared to connect TSR2 and TSR4 in an almost linear arrangement (*Figure 2A*), which is in contrast to the kinked conformation observed for FP1. The 2D classes suggested that the two monomers in FP2 are related by an approximate twofold rotation axis ($C_2$ symmetry) perpendicular to the plane of the dimer.

Whereas the FP2 2D classes above concurred with prior EM studies (*Smith et al., 1984*; *Alcorlo et al., 2013*), radically different 2D classes were also obtained when hFPNb1 was included (*Figure 2B*). In one extreme 2D class, FP2 appeared as a folded dimer adopting a double-pretzel structure with a maximum dimension of 210 Å where the two connecting arms are crossing over at TSR3, and like in FP1, the TSR2–TSR3 angle is sharply bent (*Figure 2B*). This FP2 conformation appeared to have a $C_2$ symmetry axis lying in the plane of the molecule in contrast to unbound FP2 where the apparent $C_2$ axis was perpendicular to the plane of the molecule. Other 2D classes obtained with hFPNb1 presented conformations that were intermediate between the open dimer observed in the absence of hFPNb1 and the folded dimer. A comparison of the SEC profiles supported significant conformational effects of hFPNb1 on FP2 in solution prior to EM sample preparation (*Figure 1—figure supplement 1D*). Importantly, hFPNb1 does not influence the oligomer distribution as shown previously (*Pedersen et al., 2020*).

We next analyzed the properdin trimer (FP3) by nsEM, where eight very similar 2D classes contained >90% of the particles picked (*Figure 2C*). In these classes, FP3 appeared in projection as flat triangular molecules with pseudo-$C_3$ symmetry having maximum edges of 280 Å with mainly straight connections formed by TSR2, TSR3, and TSR4 between the vertices. Again, when hFPNb1 was present, some 2D classes contained a less

planar FP3 with a maximal dimension of 240 Å where only the eye-shaped vertices could be unambiguously identified (*Figure 2D*). Other FP3-hFPNb1 2D classes featured molecules that were flat and circular like those obtained in the absence of hFPNb1 or intermediate between these two extremes. Finally, we analyzed the properdin tetramer (FP4) by nsEM and again >90% of the picked particles contributed to 2D classes with flat molecules of pseudo-$C_4$ symmetry with a maximum extent of 380 Å (*Figure 2E*). For both FP3 and FP4 2D classes, a kink was occasionally observed,

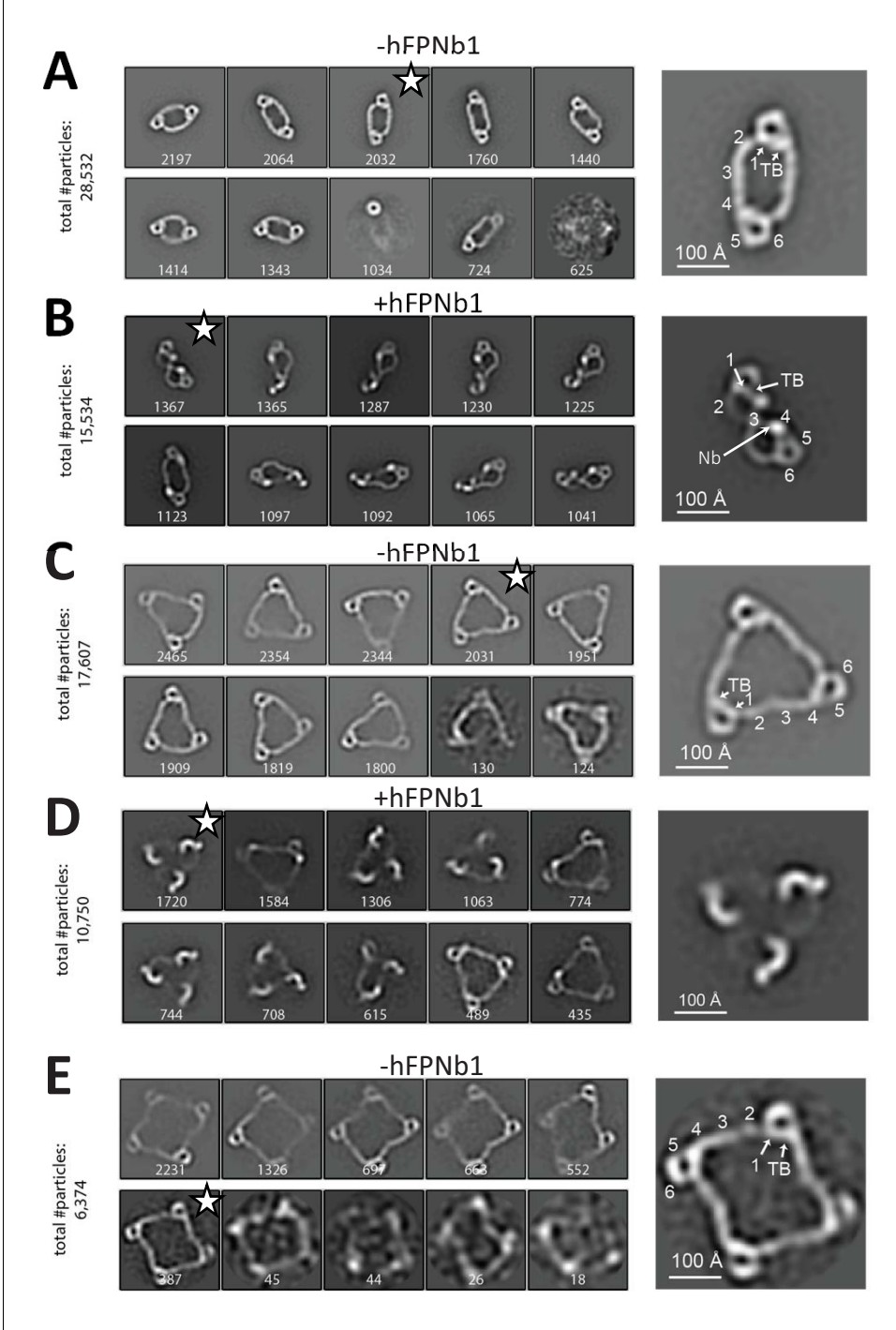

**Figure 2.** Negative-stain EM analysis of FP oligomers. For all cases, the 10 most populated nsEM 2D classes obtained are displayed with the number of particles indicated. (**A**) The 2D classes obtained with FP2 and a magnified view of the 2D class marked by star is shown to the right with the putative domain assignment indicated for one of the two monomers in the dimer. Notice the difference in curvature at TSR2 and TSR4 of the connecting arms, which facilitates the domain assignment. (**B**) As (**A**), but with the FP2-hFPNb1 complex. The double-pretzel conformation is present in two classes, while others feature the elliptical shape or intermediates. (**C**) The 2D classes obtained with FP3 reveal a flat molecule structure of apparent $C_3$ symmetry. (**D**) The FP3-hFPNb1 complex. The 2D class to the left reveals an FP3 conformation with an apparent $C_3$ symmetry that is radically different from the cyclic appearance of FP3 in (**C**), but 2D classes presenting flat cyclic FP3 and intermediates are also present. (**E**) The 2D classes obtained with FP4 suggest a planar extended molecule with apparent $C_4$ symmetry.

presumably at the TSR2–TSR3 interface, in at least one of the TSR2–TSR3–TSR4 connections. In summary, our nsEM analysis revealed unambiguous 2D classes for all the naturally occurring FP oligomers, and in all three cases, the vast majority of classified particles were present in rather similar classes representing flat molecules with apparent $C_n$ symmetry. Furthermore, the oligomer conformations observed in our 2D classes were not rare since the sum of particles used for the 2D classes for all three oligomers represented at least 50% of the particles picked (*Figure 2*). Intriguingly, we also observed that the TSR4-specific hFPNb1 could induce alternative folded conformations of the TSR2–TSR3–TSR4 arms in FP2 and FP3 that appeared to be in equilibrium through intermediates with their elliptical and flat triangular conformations.

## The solution conformation of FP oligomers

Taking an orthogonal approach to our nsEM analysis of the FP oligomers, we collected SEC-SAXS and static SAXS data for FP2, FP3, and FP4 that was SEC fractionated prior to SAXS analysis to obtain samples optimized with respect to the relevant FP oligomer (*Figure 3A–C*, *Figure 3—figure supplement 1A–G*). It is noteworthy that the SAXS data of all three oligomers exhibit characteristic bumps, which are also reflected in their $p(r)$ functions (see inserts to SAXS data and *Figure 3—figure supplement 1H*). The presence of these pronounced features clearly indicates that the oligomers are rather rigid and well defined. If conformational freedom had been present, this would smear out the SAXS data. The $p(r)$ functions of FP1, FP2, FP3, and FP4 all exhibit an initial bump at around 40 Å corresponding well to the repeated distance across the eye that is also visible in the nsEM pictures of all three oligomers (*Figures 1B* and *2*). The FP2 has a $D_{max}$ value of 230 Å. Along with a second well-defined peak at around 80 Å, which is most likely related to the distance between the two antiparallel arms connecting the two eyes, the $D_{max}$ value indicates that the FP2 solution structure is in better agreement with the extended conformation of FP2 (*Figure 2A*), than it is with the more compact twisted conformation induced by hFPNb1 (*Figure 2B*). Finally, the $p(r)$ of FP2 has a broad peak at 165 Å corresponding well to the distance between the two eyes. The $D_{max}$ of the FP3 and FP4 $p(r)$ functions are, respectively, 250 and 360 Å (*Figure 3B,C*, *Figure 3—figure supplement 1H*) and in good agreement with their larger sizes also seen in the nsEM 2D classes (*Figure 2C,E*). For the FP3 and FP4, the $p(r)$ middle peak at around 80–100 Å appears broader, less pronounced, and moves to higher values as the oligomer size increases. FP3 and FP4 have a high $p(r)$ peak at 180 and 195 Å, respectively, which is most likely the result of neighbor eye–eye distances. The increase of the eye–eye peak position when comparing FP2, FP3, and FP4 suggests that the eyes organize in a more planar structure as the oligomerization increases. Furthermore, the FP4 $p(r)$ function exhibits an additional small shoulder at 280 Å, which corresponds well to the less-frequent diagonal eye–eye distances. For a quadratic structure, as suggested by the nsEM, these would appear at $\sqrt{2}$ times the position of the neighbor eye–eye distances at 180–195 Å as they indeed do.

Using the same strategy and restraints as for FP1, we obtained rigid-body models of FP2 in the presence of $C_1$ (no) symmetry or a $C_2$ symmetry axis with $\chi^2$ in the range 1.4–3.0 for $C_1$ symmetry, whereas models with $C_2$ symmetry had $\chi^2$ of 2.4–5.5 (*Figure 3A*, *Figure 3—figure supplement 1C*). The resulting SAXS-based FP2 models all featured an extended FP2 conformation rather than the double-pretzel FP2 observed by nsEM, even if refinement was started from a double-pretzel conformation mimicking that presented in *Figure 2B*. Interestingly, the FP2 SAXS models appeared more curved as compared to the extended conformations present in EM 2D classes. In contrast to the nsEM 2D classes, the rigid-body models cannot be projected such that the two eye shapes at the opposite ends of FP2 become visible simultaneously. Hence, either rigid-body modeling could not reach the conformation observed in nsEM or the FP2 conformation is influenced by the stain and contacts with the grid and therefore become excessively flat in our 2D nsEM classes. An effect of the grid is in accordance with a lower solution $D_{max}$ compared to the maximum extent of FP2 in nsEM 2D classes.

Using the same strategy, we generated models of FP3 and FP4 by rigid-body modeling. In contrast to FP2, the $\chi^2$ values for the fit of the best models to the experimental data were comparable and in the range 1.1–1.4, indicating that the experimental data could be fitted well with single models. Importantly, the $\chi^2$ values of the output models generated with $C_3/C_4$ symmetry were similar to those generated with $C_1$ symmetry demonstrating that the fit to the data was independent of symmetry (*Figure 3B,C*, *Figure 3—figure supplement 1D,E*). All models generated with $C_1$ symmetry

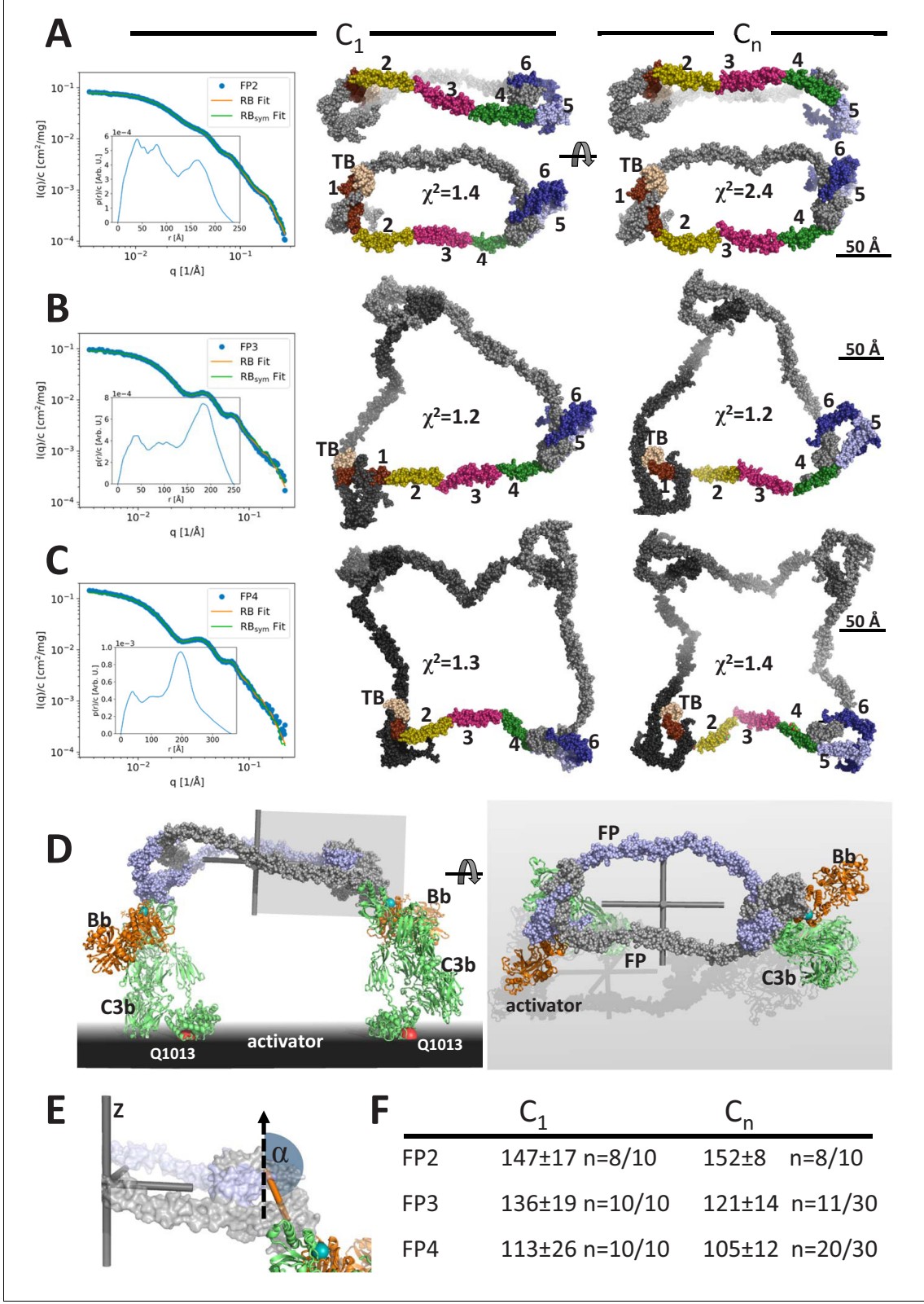

**Figure 3.** Models of FP oligomers obtained by SAXS rigid-body refinement. (**A–C**) Left: SAXS data (blue) and model fits (orange) corresponding to FP2, FP3, and FP4 structures obtained with $C_1$ symmetry. Inserts plot the corresponding $p(r)$ functions. Middle: Models fitted using $C_1$ symmetry. Right: Models fitted assuming $C_n$ symmetry. The $C_1$ models were chosen as the most representative model in the largest cluster from the 10 generated, whereas the $C_n$ models were those amongst 10 or 30 models that according to their $\chi^2$ value were in best agreement with the experimental data, see

*Figure 3 continued*

(F). In (A), two orientations are displayed to illustrate the curvature of FP2 models. (D) Hypothetical model of FP2 from panel A bound to two C3 convertases to illustrate that such a complex could be formed on an activator to which the two C3b molecules (green) carrying the Bb protease (orange) are bound through their Gln1013 (red sphere). The two FP monomers are colored light blue and gray, respectively. To the left is presented a side view and to the right a top view. The principal axes of FP2 are displayed as a gray Cartesian coordinate system. (E) Enlarged view of the area outlined with a gray box in (D). The vertical dashed vector is parallel to the smallest principal axis (labeled 'z') of the FP2 molecule, the orange stick represents the vector in the plane of the FP eye used for calculation of $\alpha$, the angle between the two vectors. (F) The average $\alpha$ values calculated from the SAXS rigid-body models. The number of models used for statistics from a pool of 10 or 30 models is indicated by the number n, remaining models were discarded as they had $\chi^2$ values that were significantly higher than the n well fitting models.

The online version of this article includes the following figure supplement(s) for figure 3:

**Figure supplement 1.** SAXS data.

were qualitatively similar extended circularized structures with their three or four eye-shaped structures joined by peripheral TSR2–TSR3–TSR4 connecting arms in agreement with the FP3 and FP4 nsEM 2D classes. As for FP2, in the models generated with $C_1$ symmetry, it was never possible to obtain projections of these SAXS models in which all eyes in FP3 and FP4 were visible. Otherwise, the SAXS models had strong resemblance to the 2D classes including an occasional kink in a TSR2–TSR3–TSR4 connection (*Figure 3B,C*). When $C_3/C_4$ symmetry was assumed during refinement, some models were trapped in a local minimum and gave rise to highly elevated $\chi^2$ values and models that were distinctly different from those with the lowest $\chi^2$ values. In contrast, the best-fitting models were all open and extended with TSR2–TSR3 at the periphery similar to those obtained with $C_1$ symmetry. For some of these models, it was possible to visualize all eyes simultaneously in projections (*Figure 3B,C*), which is in agreement with FP3 and FP4 nsEM 2D classes. Overall, our SAXS analysis of FP oligomers were in agreement with the corresponding nsEM 2D classes and suggests that the average solution conformation of FP3 and FP4 are well approximated by single models with $C_3$ and $C_4$ symmetry (*Figure 3—figure supplement 1D,E*). With respect to FP2, the situation is less clear as the experimental data fitted less well in the presence of $C_2$ symmetry, suggesting that the average FP2 conformation may be non-symmetric. Alternatively, the lower number of degrees of freedom in our rigid-body refinement performed with $C_2$ symmetry prohibits fitting the data to the extent feasible with $C_1$ symmetry and more degrees of freedom. In addition, the SAXS-based models of FP2 appear more curved than those appearing in the nsEM classes, and an effect of the EM sample preparation on the FP2 conformation cannot be excluded.

## The oligomerization interfaces can undergo very slow exchange

The oligomer distribution of FP in plasma is believed to be stable. In vitro experiments suggested that purified FP oligomers remained in their oligomerization state during storage and that when spiked into serum, radiolabeled FP oligomers did not redistribute (*Pangburn, 1989*). However, whether monomer–monomer interactions occasionally loosen up and thereby enable exchange of monomers between oligomers has never been addressed. Our prior purification of the two-chain monomeric FP molecules FPhtΔ3 and FPc (*Pedersen et al., 2019a*; *Pedersen et al., 2017*) never suggested that the two chains dissociated under native conditions. However, in size exclusion chromatography performed at low pH, it was possible to separate the two FP chains (*Figure 4A*) as previously demonstrated for FP oligomers (*Pangburn, 1989*). To investigate the stability of the FP oligomerization interfaces, we mixed our two-chain monomer FPc with the two-chain deletion mutant FPhtΔ3 monomer lacking TSR3 (*Figure 4B*). After incubation at 37°C, we purified FP molecules containing the His-tagged TSR4-6 tail fragment of FPhtΔ3. Using sodium dodecyl sulphate–polyacrylamide gel electrophoresis (SDS–PAGE) analysis, we observed that over time an increasing amount of the longer head fragment from FPc (TB-TSR1–3) co-purified with the His-tagged tail fragment from FPhtΔ3, while the amount of co-purified short FPhtΔ3 head fragment (TB-TSR1–2) decreased (*Figure 4C*). Exchange between the two FP monomers was evident after 30 min and reached equilibrium after 12 hr. To examine monomer exchange into an FP oligomer, we conducted the same experiment with FP2 and FPhtΔ3. We observed that over time, an increasing amount of full-length FP from FP2 co-purified with the FPhtΔ3 his-tagged TSR4-6 fragment, and in this case, exchange was evident after 2 hr and complete in 6–12 hr (*Figure 4D*). In conclusion, these experiments demonstrated that the oligomerization interfaces in FP can open temporarily and even

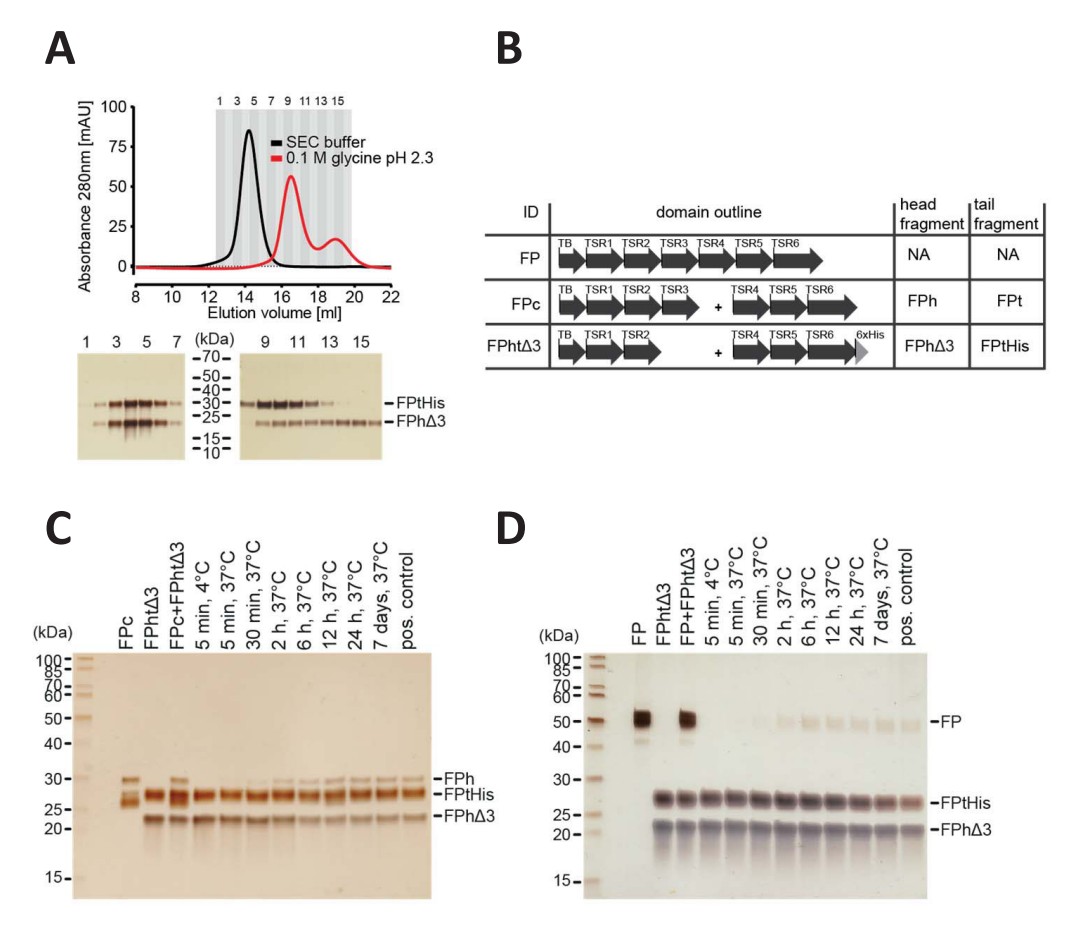

**Figure 4.** FP oligomerization interfaces can open up to enable exchange monomer. (A) SEC analysis of FPhtΔ3 in 20 mM HEPES, 150 mM NaCl, pH 7.5 (black), or 100 mM glycine pH 2.3 (red), demonstrating that the two chains of FPhtΔ3 dissociate at low pH (top). Fractions from the SEC experiments were analyzed by SDS–PAGE and silver staining showing that at neutral pH the two fragments of FPhtΔ3 co-elute (bottom, left), whereas at low pH, FPtHis (25 kDa) elute first and FPhΔ3 (18 kDa) elute last (bottom, right). (B) Overview of FP constructs used in (C) and (D). (C) A silver-stained SDS–PAGE gel following the exchange of chains between FPc and FPhtΔ3. The exchange was monitored by the appearance of the FPh chain (25 kDa) from FPc and the disappearance of the FPhΔ3 chain, after pull-down on Ni-NTA beads using the His-tag on the C-terminus of FPtHis. (D) A silver-stained gel from SDS–PAGE demonstrating the exchange of FP chains between FP2 and FPhtΔ3. The exchange was monitored by the appearance of the full-length FP (50 kDa) after pull-down on Ni-NTA beads using the C-terminal His-tag of FPtHis from FPhtΔ3.

exchange monomers with a different FP molecule under the physiologically relevant experimental conditions containing 20 mM HEPES, 150 mM NaCl pH 7.5. The exchange occurs rather slowly in our pure system, but could occur faster in a specific microenvironment in vivo. Notably, even if exchange does occur, the overall oligomer distribution may remain unaltered, and our results add further support to the concept of stable FP oligomer conformations (*Pangburn, 1989*).

## Convertase binding to FP oligomers

A major outstanding question concerning FP biology is whether the strong correlation between biological activity and oligomer size is partly due to simultaneous binding to multiple C3b molecules and convertases deposited on an activator. In FPn, there are n binding sites for C3b each located at the concave face of TSR5, and our prior structures of FP-bound C3bBb revealed that C3b binds with its major axis roughly parallel to the plane of the FP eye formed by the TB domain, TSR1, TSR5, and TSR6 (*Pedersen et al., 2019b*). Hence, as illustrated in *Figure 3D* for FP2, an oligomer with its n

convertase binding sites pointing in the same direction will have the optimal architecture for binding simultaneously to n C3b molecules deposited on an activator. Due to the overall flat shape of FP oligomers, the minor principal axis is the $C_n$ rotation axis perpendicular to the plane of the models generated with $C_n$ symmetry and a pseudo n-fold rotation axis for models generated with $C_1$ symmetry (*Figure 3E*). We could therefore quantitate the relative orientation of the convertase binding sites in our FP SAXS models by measuring the angle $\alpha$ between an appropriate vector lying in the plane of each FP eye and the smallest principal axis of the oligomer (*Figure 3E*). At the maximum value, $\alpha = 180°$, the n convertase binding sites in an FP oligomer will point in the same direction and simultaneously binding at all convertase binding sites by C3b on a planar activator appears possible. At its minimum value, $\alpha = 90°$, the convertase binding sites are parallel to the plane of the FP oligomer defined by the major and intermediate principal axes and simultaneous binding to more than two C3b molecules on a planar activator appears unlikely. We observe a clear decrease in $\alpha$ as a function of oligomer size with $\alpha \sim 150°$, 128°, and 109° for our SAXS-based models of FP2, FP3, and FP4, respectively (*Figure 3F*). A decrease in $\alpha$ with increasing multiplicity is in agreement with the shift of a major peak in the pair distance functions (*Figure 3—figure supplement 1H*) described above. This peak largely reflects the separation of neighboring FP eyes, and in curved oligomers, this separation will be smaller than in planar oligomers.

## Oligomerization alone cannot rescue FP activity

The activity of FP oligomers in assays exploring complement-dependent erythrocyte lysis follows the order FP4>FP3>FP2 (*Pangburn, 1989*), while E244K FP1 and the two-chain monomers FPc are much less active in convertase stabilization on erythrocytes and bactericidal activity compared to oligomeric FP (*Pedersen et al., 2017*). To investigate the importance of FP oligomerization and its three-dimensional structure for activity, we linked 2, 3, or 4 copies of hFPNb1 with glycine–serine linkers in expression vectors and purified the resulting hFPNb1$_2$, hFPNb1$_3$, hFPNb1$_4$. Using size exclusion chromatography, we showed that these multivalent hFPNb1 could form the expected complexes with the two-chain monomer FPc (*Figure 5A,B*). Next, we compared the activity in erythrocyte lysis in FP depleted serum of free FPc and such nanobody-linked FPc oligomers to an FP pool containing roughly equal amounts of FP2 and FP3 (*Figure 1—figure supplement 1F*). As expected, the two-chain FPc monomer required a 100-fold higher concentration compared with the FP2/FP3 oligomer pool to elicit a similar degree of lysis (*Figure 5C*). The activities of the nanobody-linked FPc oligomers were in between the activities of the FP2/FP3 pool and FPc and increased with the hFPNb1 valency of the oligomers. The erythrocyte lysis activity of these hFPNb1-linked FPc oligomers also correlated well with the dissociation of the hFPNb1-linked FPc oligomers from a C3b-coated biolayer interferometry sensor (*Figure 5D*). The FPc-hFPNb1 complex dissociated from the sensor within 75 s after transfer to the dissociation buffer. In contrast, no dissociation was observed for the FPc-hFPNb1$_4$ complex, while approx. 10% and 22% had dissociated within 75 s from the FPc-hFPNb1$_3$ and FPc-hFPNb1$_2$ complexes, respectively. In conclusion, FP activity can only be partially restored by linking FPc monomers together with the multivalent versions of hFPNb1. Importantly, these nanobody-linked FPc oligomers exhibited the anticipated slower dissociation from the C3b-coated sensor compared to the FPc monomer demonstrating that the low biological activity is not due to lack of avidity for a C3b-coated surface. Hence, the well-defined extended structure of FP oligomers demonstrated by our SAXS and EM data contributes significantly to the biological activity of FP oligomers.

## Discussion

Our demonstration of well-ordered EM 2D classes and our ability to explain solution scattering data with single models make us suggest that FP oligomers adopt a limited number of fairly stable and overall similar conformations in solution. This is a surprising finding considering that EM 2D classes of an FP oligomer have not previously been presented and the large flexibility at especially the TSR2–TSR3 and TSR3–TSR4 connections required to reach the very tight FP1 conformation (*Figure 1B–D*) as compared to FP crystal structures (*Pedersen et al., 2019a*; *van den Bos et al., 2019*). The rigidity of FP oligomers is directly manifested by the pronounced oscillations in the SAXS data leading to well-defined peaks in the pair distance distributions (*Figure 3A–C*, *Figure 3—figure*

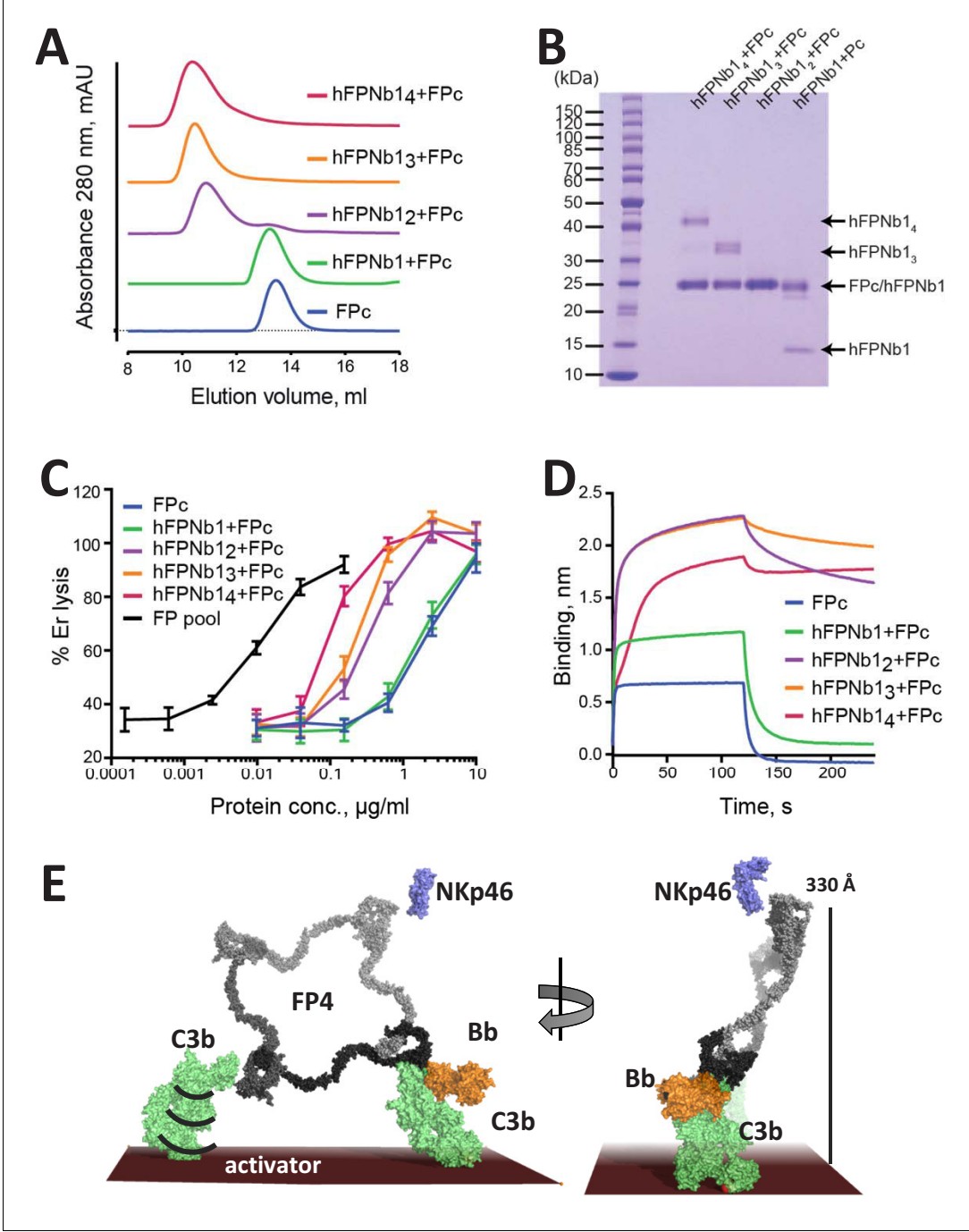

**Figure 5.** The biological activity of nanobody-linked FP oligomers is lower than that of an FP2/FP3 oligomer pool. (A) SEC profiles showing that hFPNb1 oligomers form stable oligomers with FPc. (B) SDS–PAGE analysis of the peak fraction from each SEC experiments in (A) confirming that all hFPNb1 variants form stable complexes with FPc. (C) Assay where lysis of rabbit erythrocytes (% of lysis in pure water) in FP depleted serum is used as a measure of AP activity. Erythrocyte lysis is shown as a function of protein concentration. The ability to induce erythrocyte lysis clearly increases with increasing hFPNb1-mediated oligomerization of FPc. Each point corresponds to the average of two independent experiments, both performed in technical duplicates. The error bars represent the standard deviation between the two independent experiments. (D) Biolayer interferometry sensorgrams showing the binding of FPc (blue) or preformed FPc-hFPNb1 monomers (green), dimers (purple), trimers (orange), or tetramers (red) to a C3b-coated sensor. The sensorgrams demonstrate that binding of hFPNb1-mediated FPc oligomers to C3b is stronger than for the monomeric FPc or the FPc-hFPNb1 complex. In particular, the dissociation rate is observed to be strongly dependent on hFPNb1 valency. (E) Model of FP

*Figure 5 continued on next page*

*Figure 5 continued*

structure–function relationships using FP4 as example. To the left, one convertase binding site is occupied, a neighboring binding site may form contact with a nearby C3b, proconvertase, or convertase. Flexibility of C3b due to the single-bond attachment to the activator may favor FP interaction with the second C3b molecule. Right, FP-convertase binding sites not pointing toward the activator surface reach >300 Å into solution and appear as ideal for bridging C3b-opsonized pathogens with NKp46 presenting innate lymphoid cells. Notice that the NKp46 binding site on FP is presently unknown.

The online version of this article includes the following figure supplement(s) for figure 5:

**Figure supplement 1.** Sequence alignment of FP sequences reveals a shared domain organization across major phylogenetic classes.

*supplement 1H*) that present sharp peaks between 150 and 210 Å originating from the separation of neighboring FP eyes. If the oligomers were flexible, these oscillations would be less well defined.

The solution structures of FP2 and FP3 were earlier investigated by Sun and coworkers with SAXS and analytical ultracentrifugation (*Sun et al., 2004*) and gave $R_g$ values comparable to those presented in *Figure 3—figure supplement 1A*. Their best-fitting FP2 models bear weak overall resemblance to our FP2 models, but their best-fitting FP3 model is rather different from the models we obtained. Furthermore, an extended model resembling our models of FP3 did not fit the scattering data in the study by Sun et al. One reason for these discrepancies is beyond doubt that Sun et al. did not have a detailed model of FP to base their rigid-body modeling on, and in particular they missed the crucial information regarding the stable eye structure formed by the TB domain, TSR1, TSR5, and TSR6, which we used as a single rigid body. They also used a TSR based homology model as a proxy for the N-terminal region that is a TB domain. In addition, the resolution of SAXS data in *Sun et al., 2004* was much lower with significant noise at q-values above 0.1 Å$^{-1}$, whereas we had only limited noise in the data in the range q < 0.27 Å$^{-1}$ used for rigid-body refinement. The strong preference of FP oligomers for orienting with the plane of the molecule parallel to the grid suggests that achieving high-resolution cryo-EM 3D reconstructions of these unique extended structures will be an extremely challenging task. Alternatively, FP oligomers bound to a C3b- or convertase-coated activator model may be studied with cryo-electron tomography as pioneered by Sharp and Gros for large assemblies of complement proteins (*Sharp et al., 2019*; *Sharp et al., 2016*).

The correlation between FP oligomerization and biological activity is well established and early on it was demonstrated that FP4 and FP3 bind much more avidly to C3b-coated erythrocytes compared to FP2 (*Pangburn, 1989*; *Pedersen et al., 2017*). Structural studies demonstrated that the binding site for C3b is located with an eye-shaped vertex formed by FP TSR5 and TSR6 in one FP monomer together with the TB domain and TSR1 from a second monomer (*Pedersen et al., 2019a*; *van den Bos et al., 2019*; *Alcorlo et al., 2013*). However, our functional studies revealed that FP oligomerization is not strictly required for C3b binding, stimulation of C3bB assembly, inhibition of C3bBb dissociation and competition with FI, since these activities are supported to some degree by the recombinant monomeric two-chain FPc molecule in which TSR3 and TSR4 are not connected (*Pedersen et al., 2019a*; *Pedersen et al., 2017*). The simplest explanation for the correlation between FP oligomer stoichiometry and biological activity is that of avidity due to the linking of multiple convertase binding sites and that the specific structure of FP oligomers is not important. However, such a simple model is incompatible with evidence presented in this study. We have shown that FP oligomers are structurally well ordered rather than modular structures connected by dynamic hinges (*Figures 2* and *3*). We also demonstrated that biological activity cannot be rescued solely by linking of multiple monomeric FP molecules (*Figure 5C,D*).

Our findings with respect to the role of FP oligomerization in biological activity are summarized in *Figure 5E*. Initial monovalent binding occurs when a single FP eye engages with C3b, proconvertase, or convertase. If subsequent bivalent binding occurs, it is likely to be stimulated by FB binding (*DiScipio, 1981*). The average separation of ~200 Å between convertase binding sites (*Figure 3A–D*) maintained by the rigid structure of FP oligomers also appears compatible with EM analysis of erythrocytes activated through the classic pathway that visualized tight clusters of 10–40 C3b molecules extending over 400–800 Å (*Mardiney et al., 1968*). Nevertheless, whether simultaneous interactions between oligomeric FP and multiple C3b molecules actually occur in vivo remains to be shown. Our prior SPR data showed a 450× lower apparent $K_D$ value for immobilized C3b and fluid-phase

oligomeric FP as compared to the reverse geometry, but this experiment could not distinguish between the relative contributions from multivalent binding and a high local concentration (*Pedersen et al., 2017*). In contrast, classic binding experiments measuring association with erythrocyte-bound C3bB and zymosan-C3b complexes suggested that FP oligomers bind to a C3b opsonized activator in a monovalent fashion (*DiScipio, 1981*; *Farries et al., 1988*), arguing that the local high concentration of convertase-binding sites underlies the correlation between biological activity and oligomer stoichiometry.

Although we do not know in details the oligomer distribution of FP in non-mammals, the domain structure with the TB domain followed by six thrombospondin repeats is with certainty also present in properdin sequences from amphibians, reptiles, birds, teleosts, and the agnatha *Petromyzon marinus* (*Figure 5—figure supplement 1*). This evolutionary conservation supports that a defined spatial separation of convertase-binding sites in FP oligomers is important for its function. For antibodies, it is known that the flexibility of the Fab-Fc hinge enables 'walking' over the antigen in search for bivalent attachment on spatially defined multivalent epitopes (*Zhang et al., 2020a*; *Preiner et al., 2014*). But since our results indicate that FP oligomers are quite rigid, these multivalent molecules seem not to be designed for 'walking' over the C3b-opsonized activator. Instead, the inherent flexibility of C3b and its attachment through a single covalent bond to the activator may favor multivalent FP-convertase complexes or fast rebinding through a neighboring convertase binding site (*Figure 5E*).

One reservation with respect to the model of FP structure–function relationships presented in *Figure 5E* is that our EM 2D classes obtained with the TSR4 binding hFPNb1 nanobody revealed intricate folded conformations and intermediates between the extended flat cyclic conformation observed in the absence of the nanobody (*Figure 2B,D*). This emphasizes that the FP oligomer structures we present in *Figure 3* reflect their solution conformation, but the structure may be quite different for oligomers bound to surface bound C3b and convertases on activators and NKp46 on innate lymphoid cells.

Interestingly, our structures also predict that empty C3b/convertase binding sites in FP3 and FP4 point in a direction opposite to the occupied site(s) due to their α angle being far from 180˚ (*Figures 3F* and *5E*). We estimate that such unoccupied convertase binding sites in FP3 and FP4 may protrude more than 300 Å from the thioester–activator linkage of the C3b to which an existing monovalent interaction occurs (*Figure 5E*). Such empty binding sites also appear to be suitable for bridging a C3b-opsonized cell with a second cell. Historically, FP-driven agglutination of erythrocytes was associated with large non-physiologically FP oligomers (*Farries et al., 1987*), but the contribution from each of the three naturally occurring FP oligomers to agglutination of C3b-opsonized bacteria remains unknown. Perhaps more relevant for in vivo activity, FP can also bridge C3b-opsonized bacteria with host innate lymphoid cells presenting the NKp46 receptor, an interaction shown to be required for survival in an animal model of *Neisseria meningitidis* infection (*Narni-Mancinelli et al., 2017*). Possibly, lack of this cell-bridging function is an important element in the phenotypes of FP deficiencies in addition to compromised convertase stabilization due to weaker binding to C3b on opsonized pathogens. Interestingly, administration of recombinant FP with a high content of FP tetramers and higher oligomers was protective in mouse models of infection with *N. meningitidis* and *Streptococcus pneumonia* (*Ali et al., 2014*). Possibly, the bridging of bacteria and innate lymphoid cells by larger FP oligomers contributed significantly to the beneficial effects of recombinant FP administration. Very recently, FP4, but not FP3 and FP2, was shown to act as a C3b-independent pattern recognition molecule on bacteria binding soluble collectin 12 (*Zhang et al., 2020b*), suggesting a specialized function of FP4.

Our identification of FP oligomers as rigid molecules with a potential for bridging C3b-opsonized cells with other cells may assist future studies analyzing the role of FP in complement-driven pathogenesis and facilitate new strategies for therapeutic modulation of FP activity. Inhibition of FP by function-blocking anti-mouse FP mAbs or FP gene deletion has demonstrated beneficial effects in murine models of arthritis (*Kimura et al., 2010*), renal ischemia-reperfusion injury (*Miwa et al., 2013*), allergen-induced airway inflammation (*Wang et al., 2015*), abdominal aortic aneurysm (*Bertram et al., 2015*), and atypical hemolytic uremic syndrome (*Ueda et al., 2018*; *Chen et al., 2020*). Finally, our results should promote experiments clarifying how FP oligomers may act as C3b-independent pattern recognition molecules capable of initiating the AP.

# Materials and methods

## Key resources table

| Reagent type (species) or resource | Designation | Source or reference | Identifiers | Additional information |
|---|---|---|---|---|
| Strain, strain background (*Escherichia coli*) | *E. coli* BL21(DE3) LOBSTR strain | PMID:23852738 | | hFPNb1 expression system |
| Cell line (*Homo sapiens*) | HEK293F | Thermo Fisher Scientific | R79007 | FP expression system |
| Transfected construct (*Homo sapiens*) | FP | This paper | | Construct used for expression and purification of the different FP oligomers. Construct is based on that described in PMID:28264884 and modified with a C-term His-tag |
| Transfected construct (*Homo sapiens*) | FPc | PMID:28264884 | | Monomeric FP used in size exclusion chromatography, biolayer interferometry and exchange experiments |
| Transfected construct (*Homo sapiens*) | FPthΔ3 | PMID:31507604 | | Monomeric FP lacking TSR3 used in exchange experiment |
| Biological sample (*Oryctolagus cuniculus*) | Er | Statens Serum Institut, Denmark | | Rabbit erythrocytes used in AP activity assay |
| Biological sample (*Homo sapiens*) | FP depleted serum | Complement Technologies | A339 | Human FP depleted serum |
| Antibody | hFPNb1. Lama monoclonal VHH domain. | PMID:31507604 | | Nanobody recognizing TSR4 domain of human FP used for size exclusion chromatography and negative-stain EM. |
| Commercial assay or kit | Silver Quest | Thermo Fisher Scientific | LC6070 | Silver stain kit |
| Software, algorithm | CORAL | PMID:25484842 | | |
| Software, algorithm | Gromacs 2019.4 and 2019.5 | 10.1016/j.softx.2015.06.001 | | Software used for MD simulations |
| Software, algorithm | Crysol3 | PMID:28808438 | | Calculation of theoretical SAXS curves |
| Software, algorithm | RELION | PMID:23000701 | | Data processing of negative- stain EM micrographs |
| Other | Biotinylated human C3b | PMID:28264884 | | Biotinylated C3b used for biolayer interferometry |

## Protein production and SEC assays

DNA encoding FP with a C-terminal TEV-His sequence was generated by site-directed mutagenesis and was expressed by transient expression in HEK293F cells as described in *Pedersen et al., 2017*. FP was purified from cell supernatants using a HisExcel column (GE Healthcare) and a 1 mL Mono S column (GE Healthcare) as described (*Pedersen et al., 2019b*). Fractions containing FP1, FP2, FP3, or FP4 from the Mono S column were pooled and further purified by SEC performed on a 24 mL Superdex 200 increase column (GE Healthcare) at 4°C with a flow rate of 0.25 mL/min in a buffer containing 20 mM HEPES, 150 mM NaCl pH 7.5. The monomeric FP variants FPthΔ3 and FPc used for the exchange experiments were expressed and purified as described in *Pedersen et al., 2019a*; *Pedersen et al., 2017*, respectively. FPc used in biolayer interferometry experiments and AP activity assays was expressed and purified as described in *Pedersen et al., 2017*. hFPNb1 were expressed and purified as described in *Jensen et al., 2018*. SEC analysis of FPc-hFPNb1 complexes was performed with 200 µL samples containing 14 µg FPc in complex with hFPNb1 or its multivalent derivatives using 10% molar excess of FPc compared to the number of hFPNb1 subunits. Samples were incubated in SEC buffer at room temperature for 15 min before injection.

## SAXS data acquisition, analysis, and rigid-body analysis

SAXS data was collected at the EMBL beamline P12 at PETRA III in Hamburg, Germany (*Blanchet et al., 2015*). The temperature for the sample changer and exposure unit was set to 8°C, and the detector and X-ray energy was configured to give a *q*-range of 0.0023–0.7332 Å$^{-1}$, with $q = 4\pi\sin(\theta)/\lambda$, where $\lambda$ is the wavelength of the X-ray beam and $\theta$ is the half scattering angle. These dimer data collected using the sample changer was inspected for radiation damage and averaged and background subtracted using primus in the ATSAS suite (*Franke et al., 2017*). SEC-SAXS data was collected with an in-line 24 mL Superdex 200 increase column operated at a flow rate of 0.5 mL/min. The trimer and tetramer data from SEC-SAXS were reduced using the chromixs tool from the ATSAS suite (SEC curves are presented in <u>Figure 3 - figure supplement 1</u>). Water was used as reference to convert data to absolute scale units of cm$^{-1}$ (*Orthaber et al., 2000*). Initial Guinier analysis to verify that no larger aggregates were present (data not shown) and indirect Fourier transformation to determine the *p(r)* functions were performed using the BayesApp software (*Hansen, 2012*) available through the 'GenApp.Rocks' server maintained by Emre Brookes at University of Texas. For plotting of the scattering data, the ~800 data points were logarithmically rebinned into ~180 points with better high-*q* statistics. Also, the data are only plotted in the fitted range out to about 0.25 Å$^{-1}$. Central model-independent parameters of the SAXS analysis are presented in *Figure 3—figure supplement 1A*.

Rigid-body modeling of FP monomer and oligomers was performed in CORAL (*Petoukhov et al., 2012*). The FP eye formed by the TB domain, TSR1, TSR4, TSR5, and TSR6 was used as a single rigid body. TSR2 and TSR3 from the crystal structure of FPc (entry 6RUS) formed two additional bodies that were linked to each other and the eye with distance restraints. In addition, an intact Asn-linked complex glycan (entry 3RY6) formed a fourth rigid body linked to FP Asn428, whereas mannosyl groups linked to tryptophans and fucose-glucose disaccharides linked to serine and threonine were included in the same rigid body as the TSR domain they form a covalent bond with. The starting models were constructed such that the two residues to be connected across the rigid bodies were in proximity. Starting models of FP oligomers were generated by C$_n$ symmetry and oriented with their rotation axis along the z-axis. For each FP system, six different eyes derived from entries 6S08, 6S0A, 6S0B, 6RUS, 6SEJ, and 6RUR (*Pedersen et al., 2019a*; *van den Bos et al., 2019*) were first evaluated with a consistent set of distance restraints to identify the optimal eye for the final rigid-body refinements. The α angle in *Figure 3E,F* was calculated as the angle between a vector connecting the C$_\alpha$ atoms of FP residues Ala402 and Ser345 and the shortest principal axis of the SAXS rigid-body models. If the average α was <90, the model was flipped to yield an average α > 90. Ser345 and Ala402 were chosen for definition of the FP eye vector, as their difference vector is in the plane of the FP eye and roughly parallel to the long axis of the C3b in C3bBbFP complex (*Pedersen et al., 2019a*). Scattering data together with an example of output model and fit to the experimental data are deposited in the SASBDB for the FP dimer, trimer, and tetramer. Scattering data and model for the FP E244K monomer is available as SASBDB entry SASDB69. Scattering data and models for the wild-type oligomers are available at SASBDB as entries SASDKA4 (FP2), SASDKB4 (FP3), and SASDKC4 (FP4).

## Modeling and MD simulations of the FP E244K monomer

The FP1 monomer obtained by CORAL rigid-body modeling was used as the template to construct an initial atomistic model of FP E244K with Modeller9.18 (*Webb and Sali, 2016*) in which missing residues in loops connecting domains were added and the disulfide bond Cys132–Cys170 was established. Glycosylations were added including the Asn-linked glycan at Asn428, O-linked Glucoseβ1-3Fucose at Thr92, Thr151, Ser208, and Thr272, and C-mannosylations at Trp83, Trp86, Trp139, Trp142, Trp145, Trp196, Trp199, Trp202, Trp260, Trp263, Trp321, Trp324, Trp382, Trp385, and Trp388. The glycan at Asn428 was modeled as a complex glycan. FP1 E244K was placed into a periodic cubic box with sides of 146 Å solvated with TIP3P water molecules containing Na$^+$ and Cl$^-$ ions at 0.15 M, resulting in ~300,000 atoms in total. The CHARMM36m force field (*Huang et al., 2017*) was used for the protein. Force field parameters for N- and O-linked glycans were generated using the Glycan Modeler module in the CHARMM-GUI web interface (*Park et al., 2019*). Force field parameters for C-Mannosyl Trp were obtained from *Shcherbakova et al., 2019*. Neighbor searching was performed every 20 steps. The <u>particle mesh Ewald</u> algorithm was used for electrostatic

interactions with a cut-off of 12 Å. A reciprocal grid of 128 × 128 × 128 cells was used with fourth-order B-spline interpolation. A single cut-off of 12 Å was used for Van der Waals interactions. MD simulations were performed using Gromacs 2019.4 or 2019.5 (*Abraham et al., 2015*). The temperature and pressure were kept constant at 300 K using the Nose–Hoover thermostat and at 1.0 bar using the Parrinello–Rahman barostat with a time constant of 5 ps and a frequency of 20 for coupling the pressure, respectively. Two independent MD simulations (1 µs for each) were performed to collect the conformational ensemble. These sampled conformations were used for further ensemble refinement using the BME method guided by experimental SAXS data as described (*Weinhäupl et al., 2018*; *Orioli et al., 2020*; *Bottaro et al., 2020*). By tuning the regularization parameter in the BME reweighting algorithm, we adjusted the conformational weights to various degrees to improve the fitting with the experimental SAXS data for FP E244K. VMD and PyMol were used for visualization of the conformational ensemble and movie preparation. The theoretical SAXS curve for each frame was back calculated using Crysol3 (*Franke et al., 2017*).

## Single-particle nsEM data acquisition and analysis

All samples were purified on a 24 mL Superdex 200 increase size exclusion column equilibrated in 20 mM HEPES pH 7.5, 150 mM NaCl and subsequently adsorbed to glow discharged carbon-coated copper grids, washed with deionized water, and stained with 2% (wt/vol) uranyl formate. Images were acquired with a FEI Tecnai G2 Spirit transmission microscope at 120 kV, a nominal magnification of 67.000×, and a defocus ranging from 0.7 to 1.7 µm. Automated image acquisition was performed using leginon (*Carragher et al., 2000*). For the FP1 monomer and its hFPNb1 complex, contrast transfer function (CTF) estimation and subsequent particle picking and extraction were carried out with cisTEM (*Grant et al., 2018*). For the remaining samples, CTF estimation, manual particle picking, and extraction were performed with RELION (*Scheres, 2012*). Initial 2D classes were generated and used to set up template-based particle picking. For all samples, 2D classification was performed in RELION.

## FPc and FP exchange assay

The stability of FPhtΔ3 under acidic conditions was evaluated by SEC on a 24 mL Superdex 200 column (GE Healthcare) equilibrated in 20 mM HEPES, 150 mM NaCl pH 7.5, or 100 mM glycine pH 2.3. Samples of 100 µL FPhtΔ3 at 1 mg/mL were injected and eluted at 0.5 mL/min at room temperature. Fractions were analyzed by SDS–PAGE followed by silver staining using the SilverQuest silver staining kit (Thermo Fisher). Fractions containing 0.1 M glycine pH 2.3 were neutralized with 100 µL 2 M Tris pH 8.5 before SDS–PAGE analysis. Five micrograms of FPc or dimeric FP were mixed with 5 µg of the his-tagged FPhtΔ3 in 100 µL of 20 mM HEPES, and 150 mM NaCl pH 7.5 and were incubated on ice for 5 min or at 37°C for 5 min, 30 min, 2 hr, 6 hr, 12 hr, 24 hr, or 7 days. As a positive control, 5 µg of FPc or FP was mixed with 5 µg of FPhtΔ3 in 100 µL 20 mM HEPES, 150 mM NaCl pH 7.5 and subsequently acidified by adding 100 µL 0.1 M glycine pH 2.3 to cause oligomer dissociation (*Pangburn, 1989*). The sample was incubated for 5 min at room temperature before 20 µL of 2M Tris pH 8.5 was added for neutralization. Pull-downs were performed on all samples using 50 µL of Ni-NTA beads. The beads were transferred to a 1 mL spin columns (Bio-Rad) and equilibrated in 100 mM HEPES, 0.5 M NaCl, 30 mM imidazole pH 7.5. The samples were then transferred to the columns and incubated for 2 min, followed by a 30 s centrifugation step at 70 g. The beads were washed five times with 500 µL of 100 mM HEPES, 0.5 M NaCl, 30 mM imidazole pH 7.5 before bound protein was eluted with 80 µL of 100 mM HEPES, 0.5 M NaCl, and 400 mM imidazole pH 7.5. The eluates were retrieved from the columns by a 30 s centrifugation step at 70 g. The eluates were re-applied to the column, and the centrifugation step was repeated. The samples were analyzed under non-reducing conditions on a 12% SDS–PAGE gel (GenScript) using the SilverQuest silver staining kit (Invitrogen). The oligomeric FP used in this assay was approximately 90% dimer and 10% trimer as judged by SEC analysis performed on 24 mL Superdex 200 increase column (*Figure 1—figure supplement 1G*). An SEC standard (Bio-Rad) was used for comparison.

## Biolayer interferometry assays

Biolayer interferometry experiments were performed on an Octet Red96 (ForteBio) at 30°C and shaking at 1000 RPM. Binding of hFPNb1:FPc complexes was tested on streptavidin sensor tips (SA,

ForteBio) equilibrated in assay buffer (phosphate-buffered saline supplemented with 1 mg/mL bovine serum albumin and 0.05% Tween 20) and coated with biotinylated C3b at 16 µg/mL for 10 min. FPc or hFPNb1:FPc complexes were prepared by size exclusion chromatography and diluted to 50 µg FPc/mL in assay buffer. Association was monitored for 120 s followed by 120 s dissociation in assay buffer.

## Erythrocyte lysis assay for AP activity

Rabbit erythrocytes (Er) in 11.38 mM D-glucose, 2.72 mM mono basic sodium citrate, 2.19 mM citric acid, and 7.19 mM NaCl (Alsever's solution, Statens Seruminstitut) were washed and re-suspended in AP assay buffer (5 mM barbital, 145 mM NaCl, 10 mM EGTA, 5 mM $MgCl_2$, pH 7.4, with 0.1% [wt/vol] gelatin) to obtain a 6% (vol/vol) suspension. Samples of 20 µL human FP-depleted serum diluted in assay buffer and supplemented with WT FP, FPc, or hFPNb1-FPc complexes were prepared separately and then transferred to a V-shaped bottom 96-well microtiter plate (Nunc) in duplicates. The WT FP used for this assay was approximately 50% dimer and 50% trimer as judged by SEC analysis performed on a 24 mL Superdex 200 increase (*Figure 1—figure supplement 1F*). Ten microliters of the Er suspension were then transferred to the assay plate. The plate was mixed well and incubated for 2 hr at 37°C and shaken every 30 min. Hemolysis was stopped by adding 40 µL ice-cold 0.9% NaCl, 5 mM EDTA to each well. The plate was centrifuged at 90 *g* for 10 min, and 50 µL of each supernatant was subsequently transferred to a flat-bottom microtiter well plate (Nunc). Hemolysis was then determined from the $A_{405}$ measured on a Victor3 plate reader (Perki-nElmer). Results are expressed relative to total hemolysis (obtained with water alone) and to background hemolysis (EDTA sample).

## Acknowledgements

We thank the staff at the P12 beamline at PETRAIII for help during data collection and Karen Margrethe Nielsen and Anette Hansen for technical support. We acknowledge access to computational resources from the Danish National Supercomputer for Life Sciences (Computerome) and the ROBUST Resource for Biomolecular Simulations supported by the Novo Nordisk Foundation grant no. NNF18OC0032608. This work was supported by the Lundbeck Foundation (BRAINSTRUC, grant no. R155-2015-2666) and the Novo Nordisk Foundation (NNF16OC0022058). The authors declare no conflicts of interest in relation to this manuscript.

## Additional information

### Funding

| Funder | Grant reference number | Author |
| --- | --- | --- |
| Lundbeckfonden | R155-2015-2666 | Dennis V Pedersen<br>Sofia MM Mazarakis<br>Yong Wang<br>Kresten Lindorff-Larsen<br>Lise Arleth<br>Gregers R Andersen |
| Novo Nordisk Fonden | NNF16OC0022058 | Dennis V Pedersen<br>Sofia MM Mazarakis<br>Gregers R Andersen |
| Novo Nordisk Fonden | NNF18OC0032608 | Kresten Lindorff-Larsen |

The funders had no role in study design, data collection and interpretation, or the decision to submit the work for publication.

### Author contributions

Dennis V Pedersen, Conceptualization, Resources, Data curation, Supervision, Validation, Investigation, Visualization, Methodology, Writing - original draft; Martin Nors Pedersen, Data curation, Software, Formal analysis, Validation, Investigation, Visualization, Methodology, Writing - original draft; Sofia MM Mazarakis, Resources, Validation, Investigation; Yong Wang, Data curation, Formal

analysis, Investigation, Visualization, Writing - original draft; Kresten Lindorff-Larsen, Software, Formal analysis, Supervision, Funding acquisition, Methodology, Project administration; Lise Arleth, Conceptualization, Data curation, Formal analysis, Supervision, Funding acquisition, Methodology, Writing - original draft, Project administration; Gregers R Andersen, Conceptualization, Data curation, Formal analysis, Supervision, Funding acquisition, Validation, Visualization, Methodology, Writing - original draft, Project administration, Writing - review and editing

## Author ORCIDs
Kresten Lindorff-Larsen (iD) http://orcid.org/0000-0002-4750-6039
Gregers R Andersen (iD) https://orcid.org/0000-0001-6292-3319

## Decision letter and Author response
Decision letter https://doi.org/10.7554/eLife.63356.sa1
Author response https://doi.org/10.7554/eLife.63356.sa2

## Additional files

### Supplementary files
• Transparent reporting form

### Data availability
Scattering data and model for the FP E244K monomer is available as SASBDB entry SASDB69. Scattering data and models for the wild type oligomers are available at SASBDB as entries SASDKA4 (FP2), SASDKB4 (FP3) and SASDKC4 (FP4).

The following datasets were generated:

| Author(s) | Year | Dataset title | Dataset URL | Database and Identifier |
|---|---|---|---|---|
| Andersen GR, Pedersen DV, Pedersen MN, Mazarakis SM, Wang Y, Lindorff-Larsen K, Arleth L | 2021 | Properdin oligomers adopt rigid extended conformations supporting function | https://www.sasbdb.org/data/SASDKA4/ | SASBDB, SASDKA4 |
| Andersen GR, Pedersen DV, Pedersen MN, Mazarakis SM, Wang Y, Lindorff-Larsen K, Arleth L | 2021 | Properdin oligomers adopt rigid extended conformations supporting function | https://www.sasbdb.org/data/SASDKB4/ | SASBDB, SASDKB4 |
| Andersen GR, Pedersen DV, Pedersen MN, Mazarakis SM, Wang Y, Lindorff-Larsen K, Arleth L | 2021 | Properdin oligomers adopt rigid extended conformations supporting function | https://www.sasbdb.org/data/SASDKC4/ | SASBDB, SASDKC4 |
| Andersen GR, Pedersen DV, Pedersen MN, Mazarakis SM, Wang Y, Lindorff-Larsen K, Arleth L | 2021 | Properdin oligomers adopt rigid extended conformations supporting function | https://www.sasbdb.org/data/SASDB69/ | SASBDB, SASDB69 |

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
