## [Decision Letter]

**Acceptance summary:**

This manuscript provides us with important new information on the structural features of properdin oligomers. It also gives us insight into how the higher-order structural attributes of properdin influence its function as positive regulator of complement activity. In addition to its interest to the broader immunology community, the diverse experimental approaches employed by the authors should appeal to readers from other audiences who appreciate structure-based insights.

**Decision letter after peer review:**

Thank you for submitting your article "Properdin oligomers adopt rigid extended conformations supporting function" for consideration by *eLife*. Your article has been reviewed by three peer reviewers, including Brian V Geisbrecht as the Reviewing Editor and Reviewer #1, and the evaluation has been overseen by Cynthia Wolberger as the Senior Editor. The following individual involved in review of your submission has agreed to reveal their identity: Viviana Ferreira (Reviewer #3).

The reviewers have discussed the reviews with one another and the Reviewing Editor has drafted this decision to help you prepare a revised submission.

Summary:

The manuscript under consideration has provided the following insights to our understanding of properdin structure/function:

1) Building upon previous work published by Alcorlo et al. in 2013 (Alcorlo et al., 2013), the authors use a recombinant preparation of properdin to isolate defined subpopulations of properdin oligomers and characterize these structures by a combination of negative stained electron microscopy and small-angle X-ray scattering. This work reveals that properdin oligomers are structurally rigid entities where the various oligomerization states are defined by rotational symmetry.

2) Using an immunoaffinity reagent, the authors further assign the locations of individual domains in the various oligomerization states of properdin. This allows a greater level of structural insight than is currently available.

3) The authors continue by showing that the structural rigidity of properdin oligomers impairs the extent to which additional convertase binding sites in properdin oligomers can interact with nearby convertases. This implies, but does not demonstrate directly, that these other sites may function in clustering C3b-opsonized cells for clearance.

4) Finally, the authors provide evidence that biological cross-linking of properdin alone and driving its oligomerization isn't sufficient to restore full biological activity. This result lends support to the concept that the rigid and well-defined structures of properdin oligomers are main determinants of its function.

All reviewers agree that the quality of the structural data in this manuscript are impressive. By themselves, however, these data do not expand all that much our understanding of properdin from what was presented in the Alcorlo paper from 2013. There is some sentiment that the authors overstate the novelty of this work to some degree. Recognizing this limitation, all reviewers agree that the results enumerated in points 2-4 above are important and will benefit the field moving forward. Although these points are not necessarily paradigm shifting, they collectively help further our knowledge of what remains one of the least understood components of the complement system. While each reviewer has offered some minor areas for improvement in their individual critiques, all three agree on two major points in need of clarification by revision. Since all of these items can be addressed by text revision, no further experiments are required at this point.

Essential revisions:

1) In the subsection entitled "The oligomerization interfaces…", the authors state that "these experiments for the first time demonstrated that the oligomerization interfaces in FP can open temporarily and even exchange monomers with a different FP molecule under physiologically relevant experimental conditions." However, the rearrangement of properdin oligomers was first shown by Pangburn in 1989 (Pangburn, 1989). They should remove the statement "for the first time" or re-word it. In addition, the authors state that "the exchange occurs rather slowly in our pure system and is probably not a significant reaction in the extracellular environment after secretion". This is an over-interpretation of their data because there is no way they can conclude the in vivo relevance, or lack thereof, as they cannot predict how microenvironment may affect oligomerization (this remains unknown). This sentence should be rewritten as well. Reworking this section to analyze the data at hand without undue speculation is advisable.

2) The experiments in the final subsection of the Results ("Oligomerization alone cannot rescue…."), which relied on forced oligomerization of a form of properdin that is otherwise monomeric, are not described as clearly as the remainder of the text. Given the importance of these experiments to the overall theme of this manuscript, they are not as accessible to a broad audience as they probably should be. It would be beneficial to rework this portion of the text.

---

## [Author Response]

Essential revisions:1) In the subsection entitled "The oligomerization interfaces…", the authors state that "these experiments for the first time demonstrated that the oligomerization interfaces in FP can open temporarily and even exchange monomers with a different FP molecule under physiologically relevant experimental conditions." However, the rearrangement of properdin oligomers was first shown by Pangburn in 1989 (Pangburn, 1989). They should remove the statement "for the first time" or re-word it. In addition, the authors state that "the exchange occurs rather slowly in our pure system and is probably not a significant reaction in the extracellular environment after secretion". This is an over-interpretation of their data because there is no way they can conclude the in vivo relevance, or lack thereof, as they cannot predict how microenvironment may affect oligomerization (this remains unknown). This sentence should be rewritten as well. Reworking this section to analyze the data at hand without undue speculation is advisable.

We have rewritten the final sentences in the subsection and added more information to the beginning of the subsection to clarify. We now write:

“The oligomer distribution of FP in plasma is believed to be stable. in vitro experiments suggested that purified FP oligomers remained in their oligomerization state during storage and that when spiked into serum, radiolabeled FP oligomers did not redistribute (Pangburn, 1989).”

“In conclusion, these experiments demonstrated that the oligomerization interfaces in FP can open temporarily and even exchange monomers with a different FP molecule under physiologically relevant experimental conditions. […] Notably, even if exchange does occur, the overall oligomer distribution may remain unaltered, and our results add further support to the concept of stable FP oligomer conformations (Pangburn, 1989).”

We are very much aware of the excellent JI paper by Pangburn, but we also notice that he did not observe redistribution of FP oligomers under native conditions. He does state that after denaturation and renaturation, FP returns to the normal 1:2:1 distribution, but actually no data is presented to document this finding in the JI paper. Furthermore, we demonstrated slow exchange in vitro, but exchange does not necessarily lead redistribution of oligomers, this is now clarified in the final sentence.

2) The experiments in the final subsection of the results ("Oligomerization alone cannot rescue…."), which relied on forced oligomerization of a form of properdin that is otherwise monomeric, are not described as clearly as the remainder of the text. Given the importance of these experiments to the overall theme of this manuscript, they are not as accessible to a broad audience as they probably should be. It would be beneficial to rework this portion of the text.

We have rewritten this paragraph extensively to improve its readability.